# Endothelial Dysfunction Following Enhanced TMEM16A Activity in Human Pulmonary Arteries

**DOI:** 10.3390/cells9091984

**Published:** 2020-08-28

**Authors:** Davor Skofic Maurer, Diana Zabini, Chandran Nagaraj, Neha Sharma, Miklós Lengyel, Bence M. Nagy, Saša Frank, Walter Klepetko, Elisabeth Gschwandtner, Péter Enyedi, Grazyna Kwapiszewska, Horst Olschewski, Andrea Olschewski

**Affiliations:** 1Otto Loewi Research Center, Medical University of Graz, Neue Stiftingtalstraße 6, 8010 Graz, Austria; davor.skofic-maurer@medunigraz.at (D.S.M.); d.zabini@medunigraz.at (D.Z.); Grazyna.Kwapiszewska@lvr.lbg.ac.at (G.K.); 2Ludwig Boltzmann Institute for Lung Vascular Research, Neue Stiftingtalstraße 6, 8010 Graz, Austria; Nagaraj.Chandran@lvr.lbg.ac.at (C.N.); Bence.Nagy@lvr.lbg.ac.at (B.M.N.); horst.olschewski@klinikum-graz.at (H.O.); 3Experimental Anaesthesiology, Department of Anaesthesiology and Intensive Care Medicine, Medical University of Graz, Auenbruggerplatz 5, 8036 Graz, Austria; neha.sharma@medunigraz.at; 4Department of Physiology, Semmelweis University, Tűzoltó utca 37-47, 1094 Budapest, Hungary; lengyel.miklos@med.semmelweis-univ.hu (M.L.); enyedi.peter@med.semmelweis-univ.hu (P.E.); 5Gottfried Schatz Research Center, Medical University of Graz, Neue Stiftingtalstraße 6, 8010 Graz, Austria; sasa.frank@medunigraz.at; 6Department of Thoracic Surgery, Medical University of Vienna, Währinger Gürtel 18-20, 1090 Vienna, Austria; walter.klepetko@meduniwien.ac.at (W.K.); elisabeth.gschwandtner@meduniwien.ac.at (E.G.); 7Department of Internal Medicine, Division of Pulmonology, Medical University of Graz, Neue Stiftingtalstraße 6, 8010 Graz, Austria

**Keywords:** TMEM16A, Ano1, pulmonary endothelial cells, endothelial dysfunction, pulmonary hypertension, intracellular calcium, angiogenesis, eNOS uncoupling, benzbromarone, metabolic switch

## Abstract

Endothelial dysfunction is one of the hallmarks of different vascular diseases, including pulmonary arterial hypertension (PAH). Ion channelome changes have long been connected to vascular remodeling in PAH, yet only recently has the focus shifted towards Ca^2+^-activated Cl^−^ channels (CaCC). The most prominent member of the CaCC TMEM16A has been shown to contribute to the pathogenesis of idiopathic PAH (IPAH) in pulmonary arterial smooth muscle cells, however its role in the homeostasis of healthy human pulmonary arterial endothelial cells (PAECs) and in the development of endothelial dysfunction remains underrepresented. Here we report enhanced TMEM16A activity in IPAH PAECs by whole-cell patch-clamp recordings. Using adenoviral-mediated TMEM16A increase in healthy primary human PAECs in vitro and in human pulmonary arteries ex vivo, we demonstrate the functional consequences of the augmented TMEM16A activity: alterations of Ca^2+^ dynamics and eNOS activity as well as decreased NO production, PAECs proliferation, wound healing, tube formation and acetylcholine-mediated relaxation of human pulmonary arteries. We propose that the ERK1/2 pathway is specifically affected by elevated TMEM16A activity, leading to these pathological changes. With this work we introduce increased TMEM16A activity in the cell membrane of human PAECs for the development of endothelial dysfunction in PAH.

## 1. Introduction

Endothelial cells form a barrier, effectively separating the luminal from the abluminal side of the vessels and responding to a variety of cues coming from continuous intercellular communication, both locally and systemically. They also function as an endocrine organ, releasing a diverse array of compounds affecting vasoactive, immune, growth and coagulation processes [1,2]. This makes the endothelium in pulmonary arterial hypertension (PAH) a perfectly positioned critical source of mediators promoting vascular remodeling, e.g., serotonin (5-HT), vasoactive peptides (NO, PGI2, ET-1), cytokines, chemokines, and growth factors [3]. Endothelial dysfunction is one of the key players in the development of PAH, a multifactorial disorder characterized by a progressive rise in vascular resistance [3,4].

Channelopathy in PAH has been traditionally structured around the effect of K^+^ and Ca^2+^ channel alterations on the phenotype of pulmonary arterial smooth muscle cells (PASMCs) [5]. In contrast, our knowledge about aberrations in ion channel function in endothelial cells of pulmonary arteries (PAECs) is limited. Posttranslational modifications and decreased K^+^ channel expression have been described in idiopathic PAH (IPAH) PASMCs and animal models of PH, linking membrane depolarization to opening of voltage-gated Ca^2+^ channels (VGCC), thus promoting the constrictive and proliferative phenotype as seen in PAH [6,7,8,9]. Moreover, transient receptor potential (TRP) channels were likewise shown to be dysregulated in IPAH PASMCs demonstrating their role in the phenotypic switch [10]. The focus has recently been extended towards Ca^2+^-activated Cl^−^ channels (CaCC), demonstrating their importance in cell physiology [11]. Since the discovery of CaCC in 2008 [12], its most prominent member TMEM16A has been shown to be a key component in maintaining ionic homeostasis of several cells and tissues, including the gut [13], airway epithelium [14], kidney [15] and brain [16]. On the other hand, its dysfunction has been associated with the pathophysiology of several heterogeneous diseases such as cancer, hypertension, gastrointestinal motility disorders and cystic fibrosis [17]. Previously we reported the contribution of TMEM16A in the maintenance of PASMC membrane potential [18]. IPAH-associated increase in the channel activity caused severe dysregulation of the Ca^2+^ homeostasis, promoting PASMC constriction and a proliferative phenotype. By using the TMEM16A inhibitor benzbromarone (Bbr) we effectively counteracted the two major hallmarks of PAH; chronic vasoconstriction and remodeling in vivo [18].

Recently increased TMEM16A activity in the mitochondria has also been shown in PAECs in IPAH [19], yet studies addressing its functional significance in the endothelial cells remain rare. Here we show the pathological consequences of increased TMEM16A activity in primary human PAECs. We introduce TMEM16A as a player responsible for pathological changes in IPAH PAECs.

## 2. Materials and Methods

### 2.1. Human Lung Samples

Human lung tissue samples were obtained from donors or patients with IPAH who underwent lung transplantation at the Department of Surgery, Division of Thoracic Surgery, Medical University of Vienna, Austria. The protocol and tissue usage were approved by the institutional ethics committee (976/2010) and patient consent was obtained before lung transplantation. The patient characteristics included: age at the time of transplantation, weight, height, sex, mean pulmonary arterial pressure (mPAP) measured by right heart catheterization, pulmonary function test, as well as oxygen and medication before transplantation. The chest computed tomography (CT) scans were reviewed and the diagnoses were verified by an independent board including experienced pathologists, radiologists and pulmonologists. Healthy donor lung tissue was obtained from the same source.

Detailed patient characteristics can be found in Appendix A.

### 2.2. Cell Isolation and Culture

#### 2.2.1. Donor and IPAH PAECs 

For the isolation of donor and IPAH pulmonary artery endothelial cells (PAECs), pulmonary arteries (<2 mm in diameter) were isolated and the endothelium incubated with an enzymatic mixture of collagenase, DNAse and dispaze in HBSS at room temperature (RT). Cell suspension was collected, resuspended in VascuLife Complete SMC Medium and cultured in gelatin-coated T25 flasks at 37 °C and 5% CO_2_. After reaching 70–80% confluency, cells were trypsinized, enriched by 3 consecutive steps of CD31-selective magnetic-activated cell sorting technology and verified via morphological and marker confirmation (smooth muscle actin, fibronectin, vimentin, von Willebrand Factor, smooth muscle myosin heavy chain and CD31). Surplus PAECs were frozen (endothelial cell complete medium containing 12% FCS and 10% DMSO) and stored in liquid nitrogen until further use. Passages 3–9 were used for the experiments.

Human PAECs purchased (Lonza, Basel, Switzerland) or isolated as described above, were cultured in gelatin-(0.1% gelatin solution in PBS) coated cell culture flasks in Lonza endothelial cell growth medium (EBMTM-2 supplemented with EGMTM-2 containing growth factors, cytokines and other supplements, with final 2% FCS concentration), here referred to as complete medium. Whenever the starvation medium was used, EBMTM-2 was supplemented with 0.5% FCS and 0.2% antibiotics.

Detailed patient characteristics of isolated and purchased PAECs can be found in Appendix A respectively.

#### 2.2.2. PASMCs

The isolation and culture of human PASMCs was performed according to Stulnig et al. [20]. After the removal of the endothelial cell layer, the media was peeled away from the underlying adventitial layer and cut into approximately 1–2 mm^2^ sections, centrifuged and resuspended in VascuLife Complete SMC Medium supplemented with 20% FCS and 0.2% antibiotics, then transferred to T75 flasks and cultured at 37 °C and 5% CO_2_. After a confluent monolayer of PASMC had formed, the cells were trypsinized and either cultured in VascuLife Complete SMC medium supplemented with 10% FCS and 0.2% antibiotics, or frozen (VascuLife Complete SMC Medium containing 15% FCS and 10% DMSO) and stored in liquid nitrogen until further use. Passages 4–8 were used for the experiments.

Detailed patient characteristics of isolated PASMCs can be found in Appendix A.

### 2.3. Precision-Cut Lung Slices (PCLS)

Donor lung cuts were filled with 3% low melting agarose and let to solidify at 4 °C for 15 min. Cylindrical cores of 8 mm in diameter were cut and sliced in cutting solution to sections of 250 μm thickness using a Krumdieck live tissue microtome. Freshly cut slices were then transferred to the incubation medium and kept at 37 °C and 5% CO_2_ in the incubator. The incubation medium was changed 4 times separated by 30 min wash steps and finally left overnight. On the following day the slices were either fixed in 4% formaldehyde for 1 h or kept for further use in culture.

Further information regarding solutions and materials can be found in Appendix A respectively.

### 2.4. Overexpression of TMEM16A

#### 2.4.1. Cell Preparation

For TMEM16A overexpression, human PAECs or PASMCs were seeded on gelatin-coated chamber slides or in 6-well plates and left to settle in fully supplemented medium overnight. On the next day the medium was replaced by fresh complete medium containing either TMEM16A-encoding adenoviruses Ano1^Ad^ or control Ctrl^Ad^ (Cyagen Biosciences, Santa Clara, CA, USA) at multiplicity of infection 50 (MOI 50). After 24 h the medium was replaced by fresh complete medium. For further assays, cells were collected either 48 or 72 h after the start of infection. 

#### 2.4.2. Isolated Pulmonary Arteries and Precision-Cut Lung Slices (PCLS)

In the case of isolated human pulmonary arteries or PCLS, the vessels or PCLS slices were incubated with the adenovirus in basal VascuLife Medium for 24 h followed by exchange of the viral solution and incubation for further 24 h. After 48 h the vessels were either used for isometric tension measurements, immunofluorescence imaging or collected for protein assessment followed by western blot.

Further information regarding solutions and materials can be found in Appendix A respectively.

### 2.5. Immunofluorescence Staining

#### 2.5.1. Human Lung sections

Formalin-fixed paraffin-embedded human lung tissue blocks were cut to 3.5 μm thick slices. Sections were deparaffinized at 60 °C overnight and antigen retrieval was performed with Dako Target Retrieval Solution pH 9.0 at 95 °C. Lung sections were blocked with 10% BSA for 1 h at RT and immunolabelled with antibodies against Von Willebrand factor (vWF) and TMEM16A at 4 °C overnight. On the following day, the sections were washed, then labelled with Alexa Fluor 555 a-rabbit and Alexa Fluor 647 a-mouse secondary antibodies for 1 h at RT. Finally, the slides were preserved using a mounting medium containing 4′,6-diamidino-2-phenylindole dihydrochloride (DAPI) and imaged with Zeiss LSM 510 META laser scanning confocal system. Duplicates were processed either without the primary antibody or with the primary antibody against TMEM16A pre-incubated with the immunogen peptide as negative controls.

#### 2.5.2. Precision-Cut Lung Slices (PCLS)

Formalin-fixed pieces were blocked at 4 °C overnight in PBS containing 5% BSA and 0.5% Triton X-100. On the next day, the slices were transferred to primary antibodies against vWF and TMEM16A prepared in the same solution and incubated for 24 h at 4 °C. On the following day, the pieces were washed, then labelled with a mixture of AlexaFluor 555 a-rabbit secondary antibody and DAPI prepared in PBS with 3% BSA and incubated for another 24 h at 4 °C. The pieces were preserved using a DAKO mounting medium and imaged with Nikon’s A1+ confocal laser microscope system.

In the case of TMEM16A overexpression, the staining procedure was similar to the one described above; the pieces were incubated with a mixture of AlexaFluor 488 a-rabbit secondary antibody and DAPI. vWF antibodies were labelled with Mix-n-Stain™ CF™ 633 antibody labeling kit according to the manufacturer’s instructions.

#### 2.5.3. Human PAECs

Cells were seeded onto gelatin-coated chamber slides and formalin-fixed. After blocking with 5% BSA, the cells were incubated overnight at 4 °C with antibodies against Von Willebrand factor and TMEM16A. On the following day, the cells were washed, then labelled with Alexa Fluor 555 a-rabbit and Alexa Fluor 647 a-mouse secondary antibodies for 1 h at RT. Finally, the slides were preserved using a mounting medium containing DAPI and imaged with Nikon’s A1+ confocal laser microscope system. Duplicates were processed either without the primary antibody or with the primary antibody against TMEM16A pre-incubated with the immunogen peptide as negative controls.

For the labelling of TMEM16A-overexpressing human PAECs, cells were infected with a MOI 50 as described above. 72 h after adenoviral infection, the cells were fixed with 4% paraformaldehyde. The following staining protocol was similar as described above; the slides were incubated with Alexa Fluor 647 a-rabbit secondary antibody and imaged with Nikon’s A1+ confocal laser microscope system.

Further information regarding antibodies and materials can be found in Appendix A respectively.

### 2.6. Analysis of Protein Expression

For the analysis of total protein, cells were lysed in CHAPS [2] buffer and analyzed with Western blot. Alternatively, pieces of healthy human lung tissue were collected in CHAPS [2] buffer followed by homogenization with beads and MagNA Lyser instrument or sonication respectively. After centrifugation (13,000 g, 10 min), the protein concentration of the supernatant was determined with Pierce BCA protein assay kit according to the manufacturer’s instruction. A specific amount of protein from each sample was mixed with 10× Loading buffer and run on 8% or 15% SDS-polyacrylamide gels, followed by electro transfer to a nitrocellulose membrane. After blocking the membrane with 5% BSA in TBS-T, the membrane was probed with one of the following antibodies: TMEM16A, pp38, p38, pERK, ERK, pAkt, pAkt, tAKT, pSAPK/JNK, SAPK/JNK, PCNA, Cl. PARP, Cyclin D1, eNOS, LC3B, pSer1177 eNOS or pThr495 eNOS. Afterwards the membrane was incubated with horseradish peroxidase conjugated secondary antibody and the final detection of the proteins was performed using a SuperSignal West Femto, ECL prime or ECL Start Kit and a ChemiDocTM touch imaging system. Protein density was normalized to the intensity of housekeeping protein. As housekeeping genes B-Actin or Vinculin were used.

For testing the effect of TMEM16A-overexpression on eNOS phosphorylation, 48 h after adenoviral infection, the cells were 2 h serum-starved and stimulated with 5 µm acetylcholine (ACh). Protein was collected at 0, 5, 15 and 30 min post-stimulation.

For testing the effect of reduced [Cl^−^]_in_ on eNOS phosphorylation, the cells were incubated for 24 h with either Ringer’s solution containing physiological [Cl^−^] or solution with KCl and NaCl exchanged with potassium and sodium gluconate to half the amount respectively ([Cl^−^] was reduced from 129.5 to 67.25 mm). After 2 h serum starvation, the rest of the protocol was performed as described above.

Further information regarding antibodies, solutions and materials can be found in Appendix A respectively.

### 2.7. Measurement of Whole-Cell Ca^2+^-Activated Cl^−^ Current (I_ClCa_)

Whole-cell Ca^2+^-activated Cl^−^ current was measured as reported previously [18]. Briefly, donor or IPAH PAECs grown on gelatin-coated dishes were harvested with StemPro Accutase, centrifuged (300 g, 10 min), resuspended in complete medium and incubated at 37 °C for 15 min to allow them to attach before measurements. In the case of adenoviral manipulation of TMEM16A expression, cells were incubated with Ano1^Ad^ or its control Ctrl^Ad^ at MOI 50 as described above (see Overexpression of TMEM16A) and used after 48 h.

The cells were incubated in bath solution I until the formation of whole-cell configuration. Once the amplitude of TMEM16A current was stable, the cells were perfused with bath solution II, containing TEA-Cl to minimize K^+^ current contamination, and vehicle or benzbromarone (Bbr). In order to measure I_ClCa_, the command potential was stepped from a 0 mV holding potential to −40, 0, +40, +80 and +120 mV for 1.5 s, allowing 0.5 s recovery time at the holding potential between each step. The average current measured between 1000 and 1500 ms of each voltage step was plotted against the holding potential. Due to the almost symmetrical Cl^−^ concentration of the bath and pipette solutions, the reversal potential (E_rev_) for Cl^−^ was expected to be around zero.

Patch pipettes were pulled from thick-walled borosilicate glass by a P-87 puller (Sutter Instrument Co., Novato, CA, USA) and fire-polished. Pipettes were filled with pipette solution (pipette resistance was between 3–5 MΩ) and connected to the headstage of an Axopatch-1D patch clamp amplifier (Axon Instruments, Inc., Foster City, CA, USA). Cut-off frequency of the eight-pole Bessel filter was adjusted to 200 Hz, data were acquired at 2 kHz. Experiments were carried out at RT (21 °C). Solutions were applied using a gravity-driven perfusion system. Data were digitally sampled by Digidata 1550B (Axon Instruments, Inc., Foster City, CA, USA) and analyzed by pCLAMP 10 software.

Further information regarding solutions and materials can be found in Appendix A respectively.

### 2.8. Live Cell Ca^2+^ Imaging

Live cell Ca^2+^-imaging was done as previously reported [21]. Human PAECs or PASMCs were seeded on 25 mm glass coverslips and infected with a viral solution of Ano1^Ad^ or Ctrl^Ad^ at MOI 50. Forty-eight hours after adenoviral infection, the cells were loaded with 2 µm Fura-2-acetoxymethyl ester (Fura-2AM) for 45 min at 37 °C. The single glass cover slip was mounted on the stage of a Zeiss 200 M inverted epifluorescence microscope coupled to a PolyChrome V monochromator (Till Photonics, Kaufbeuren, Germany) light source in a sealed, temperature-controlled RC-21B imaging chamber (Warner Instruments, Hamden, CT, USA). Fluorescence images were obtained every 3 s with alternate excitation at 340 and 380 nm and the emitted light collected at 510 nm by an air-cooled Andor Ixon camera (Andor Technology, Belfast, Ireland). Background fluorescence was recorded from each cover slip and subtracted before calculation. The acquired images were stored and processed offline with TillVision software (Till Photonics, Germany).

All solutions were prewarmed to 30 °C and cells were stimulated with a 15 µm selective SERCA blocker 2,5-Di-t-butyl-1,4-benzohydroquinone (BHQ) in the presence and absence of extracellular Ca^2+^. Maximal and minimal ratio values were determined at the end of each experiment by treating the cells with 5 µm ionomycin (maximal ratio) followed by 20 mm EGTA-mediated chelation of all free Ca^2+^ (minimal ratio). Cells that did not respond to ionomycin were discarded. The basal Ca^2+^ levels were determined as an average of initial 50 s and [Ca^2+^]_i_ was calculated as previously described [22]. BHQ-induced Ca^2+^ peak and Ca^2+^ response following Ca^2+^ readmission were calculated as the maximal peak height subtracting the baseline.

Further information regarding solutions and materials can be found in Appendix A and Appendix A respectively.

### 2.9. DAF-DM-Mediated Nitric Oxide Measurement

PAECs were seeded in gelatin-coated dark 96-well plates and treated either with TMEM16A-overexpressing adenovirus or Ringer’s solution with reduced Cl^−^.

In the case of adenovirus, following infection at MOI 50 and 48 h of overexpression, the cells were starved for 2 h with Ringer’s solution and loaded with 10 µm 4-Amino-5-Methylamino-2′,7′-Difluorofluorescein Diacetate (DAF-FM) for 30 min at 37 °C. The cells were stimulated with 5 µM ACh for 20 min followed by measurement on CLARIOstar Plus (BMG Labtech, Ortenberg, Germany) at Ex/Em = 495/515 nm. All the assays were performed in quadruplicate and normalized to protein content.

Further information regarding solutions and materials can be found in Appendix A respectively.

### 2.10. Wound Healing Assay

Human PAECs were seeded in 2-well culture-inserts (Ibidi, Planegg, Germany) and infected with either Ano1^Ad^ or Ctrl^Ad^ at MOI 50. Following overnight starvation and altogether 48 h of TMEM16A overexpression, the inserts were removed to create a gap of approximately 500 µm and the cells were immersed in a complete medium. The closing gaps were photographed at 4x magnification (Olympus CKX41) at 0, 24, 36 and 48 h after removal of the inserts and the photos quantitatively assessed comparing the initial gap with the area of the healing wound using image analyzing software (ImageJ 1.46r).

Further information regarding solutions and materials can be found in Appendix A.

### 2.11. Pulmonary Arterial Isometric Tension Measurements

All animal studies were approved by the Austrian Ministry of Education, Science and Culture under the license number BMWFW-66.010/0043-WF/V/3b/2016. Ten-to-twelve-week-old wild-type male mice were sacrificed by cervical dislocation and lungs collected for pulmonary artery (PA) isolation. Using a stereo zoom microscope SZX7 (Olympus, Tokyo, Japan) second order PAs of 4 mm in length were isolated and cut in half. One half was always used for a vehicle control. Alternatively, human PAs (4 mm), isolated from human lung tissue obtained from Vienna (976/2010), were isolated, cut in half and incubated with either Ano1^Ad^ or Ctrl^Ad^ for 24 h, after which the solution was exchanged for VascuLife SMC medium without any growth factors or FCS (LifeLine Cell Technology) for a further 24 h.

In both cases the isolated PAs were mounted on the wire myography system (Danish Myo Technology 620M, Aarhus, Denmark) with the help of tungsten wires as described previously [23]. Afterwards the PAs were equilibrated for 30 min in physiological salt solution (PSS; pH 7.4, 100% oxygen, 37 °C) followed by an increase of basal tension to 2 mN and stabilization for further 30 min. The vessel viability was tested with 3 sequential steps of depolarization, PSS with 120 mM KCl (KPSS; isotonic replacement of NaCl by KCl) each lasting 15 min. The vessels with mean KPSS response below 2 mN were discarded. The effect of vasoactive agents (high potassium chloride, thromboxane analog U44619 (300 nm), L-NAME (300 µm), Bbr (0.1–30 µm) or ACh (0.1–30 µm)) was tested by directly adding the agents into the chamber and the differences were measured between the vessels incubated with the substances or vehicle, or in the case of human PAs between TMEM16A-overexpressing Ano1^Ad^ or Ctrl^Ad^ expressing vessels.

The myography chambers were connected to force transducer units for isometric measurements (PowerLab, ADInstrument, Oxford, UK). The vasorelaxation was calculated as percentage (% relaxation) of the contraction induced by 300 nm U46619. U46619 caused a constriction of 3.88 ± 0.38, 3.99 ± 0.32 or 3.40 ± 0.20 mN in the case of isometric tension measurements of control, endothelium-removed or L-NAME-incubated mouse pulmonary arteries respectively. In the case of isometric tension measurements of human pulmonary arteries U46619 caused a constriction of 5.90 ± 1.63 or 5.90 ± 1.57 mN of control or TMEM16A-overexpressing pulmonary arteries respectively.

Further information regarding solutions and materials can be found in Appendix A respectively.

### 2.12. Measuring Cell Metabolic State

The Agilent Seahorse XFp setup (Agilent, Santa Clara, CA, USA) was used to assess the mitochondrial function of primary human PAECs: 20,000 cells were seeded into 8-well plates and infected with either Ano^Ad^ or Ctrl^Ad^ and incubated for 24 h. Forty-eight hours after adenoviral infection, a Seahorse XFp cell mito stress test kit was used according to the manufacturer’s instructions. Briefly, with the use of Oligomycin, Rotenone and Rotenone/Antimycin A, the mitochondrial state can be assessed. All the assays were performed in triplicate and normalized to protein content.

Further information regarding materials can be found in Appendix A.

### 2.13. Matrigel Tube-Formation Assay

To test endothelial cell angiogenic potential, Matrigel tube formation assay was used (Merck Millipore, Burlington, MA, USA): 200,000 cells were seeded in a gelatin-covered 6-well plate and infected with either Ano1^Ad^ or Ctrl^Ad^ at MOI 50 and incubated for 24 h. Following overnight starvation and 48 h after adenoviral infection, the cells were trypsinized, counted and 50,000 cells per 96-well plate were seeded onto prepared matrigel in triplicate according to the manufacturer’s instructions. After 6 h of incubation period at 37 °C, the tubular networks were photographed at 4x magnification (Olympus CKX41) and photos quantitatively assessed comparing the number of branching points using image analyzing software (ImageJ 1.46r).

Further information regarding materials can be found in Appendix A.

### 2.14. Assessment of Cell Proliferation In Vitro

To study the influence of TMEM16A on human PAEC proliferation, 10,000 cells per well were seeded in 96-well plates and infected with either Ano1^Ad^ or Ctrl^Ad^ at MOI 50. After 24 h the medium was replaced by fresh complete medium and the cells serum-starved overnight. On the following day the cell medium was replaced by fresh starvation medium with added ^3^H-thymidine. The proliferation was determined after 24 h ^3^H-thymidine incorporation (BIOTREND Chemikalien GmbH, Cologne, Germany) and altogether 72 h after adenoviral infection as an index of DNA synthesis, and detected with a scintillation counter and the results are represented as counts per minute (CPM) (Wallac 1450 MicroBeta TriLux liquid scintillation counter and luminometer, PerkinElmer, Waltham, MA, USA). Each independent experiment was performed in quintuplicate.

Further information regarding materials can be found in Appendix A.

### 2.15. Assessment of Cell Resting Membrane Potential In Vitro

To study the influence of TMEM16A on human PAEC resting membrane potential, 80,000 cells per well in a 6-well plate were seeded and transfected with either Ctrl^Ad^ or Ano1^Ad^ at MOI 50. After 24 h the medium was replaced by fresh complete medium and the cells were serum-starved overnight. 48 h after adenoviral infection, the medium was replaced with starvation medium supplemented with 10 µm DiBAC_4_(3) dye for 30 min at 37 °C after which the cells were collected with cell-scratcher in PSS. Fluorescence signal intensity was measured on CytoFLEX S flow cytometer (Beckman Coulter, Brea, CA, USA) at Ex/Em = 490/516 nm. Increased depolarization results in additional influx of the anionic dye and an increase in fluorescence.

Further information regarding solutions and materials can be found in Appendix A respectively.

### 2.16. Assessment of Cell Cas3/Cas7 Activation In Vitro

To study the influence of TMEM16A on human PAEC Cas3/Cas7 activation, we seeded 150,000 cells per well in a 6-well plate and transfected the next day with either Ctrl^Ad^ or Ano1^Ad^ at MOI 50. After 24 h the medium was replaced by fresh complete medium and the cells serum-starved overnight. The next day the medium was exchanged by fresh starvation medium. As a positive control, we incubated the cells with 10 µm Staurosporin (STS) for 24 h. Seventy-two hours after adenoviral infection, CellEventTM Caspase-3/7 green detection reagent was used according to the manufacturer’s instruction. The fluorescent signal shift was measured with a CytoFLEX S flow cytometer (Beckman Coulter, Brea, CA, USA) at Ex/Em = 503/530 nm.

Further information regarding solutions and materials can be found in Appendix A.

### 2.17. Cell-Cycle Analysis

To study the effect of TMEM16A overexpression on cell cycle progression of primary human PAECs we seeded 200,000 cells per well in a 6-well plate and transfected with either Ctrl^Ad^ or Ano1^Ad^ at MOI 50. After 24 h the medium was replaced by fresh complete medium and the cells serum-starved overnight. The next day the medium was exchanged by fresh starvation medium for a further 24 h. Seventy-two hours after adenoviral infection, the cells were trypsinized, resuspended in PBS, then transferred into cold EtOH under continuous vortexing and incubated at 4 °C for at least 30 min. After washing using PBS supplemented with 0.5% FCS, the cell pellet was resuspended in a solution containing 1 µg/mL DAPI and 0.1% Triton x-100 in PBS and incubated at RT for 10 min, then transferred on ice until measurements were done. Fluorescence signal intensity was measured on a CytoFLEX LX flow cytometer (Beckman Coulter, Brea, CA, USA) at Ex/Em = 355/461 (bandpass filter 450/45). The data was analyzed using ModFit software.

Further information regarding materials can be found in Appendix A.

### 2.18. qRT-PCR

qRT-PCR was performed as described previously [18]. Briefly, PAECs were grown until confluence and serum-starved overnight. RNA was collected using the PeqGOLD Total RNA kit (PeqLab, Erlangen, Germany) and transcribed into cDNA with the iScript reagent mix (Bio-Rad, Hercules CA, USA). To assess ANO1 expression, exon-exon junction spanning primers targeting the boundaries of exons 1 and 2 were acquired from Eurofins, Graz, Austria (see Appendix A). The primers covered the region without reported alternative splicing, therefore they amplified all splice variants. qRT-PCR was performed using Blue S’Green qPCR kit.

Further information regarding primer sequences and materials can be found in Appendix A respectively.

### 2.19. Statistical Analysis

Data is shown either as individual data plots with median, as mean ± s.e.m or as floating bars plot (min-to-max). In all cases, n numbers are representing number of replicates, N numbers are representing number of patients. Numbers are given in the corresponding figure legend. Statistical analyses were performed using Prism 8.0 (GraphPad Software, La Jolla, CA, USA) and are identified in the corresponding figure legend. All datasets met the assumptions of the statistical test used, statistical analyses were two-sided and *p* values < 0.05 were considered significant.

## 3. Results

### 3.1. TMEM16A Shapes Pulmonary Arterial Vascular Tone

Western blots detecting TMEM16A in PAECs, pulmonary arterial smooth muscle cells (PASMCs), as well as in human lung homogenate (hLH), verified TMEM16A expression in PAECs (Figure 1a, Appendix A and Appendix A). Whole-cell voltage clamp measurements confirmed a functional Bbr-sensitive Ca^2+^-activated Cl^−^ current (I_ClCa_) in PAECs (Figure 1b). The Bbr-sensitive current was larger in human PAECs in comparison to PASMCs. Our results suggest that TMEM16A defines I_ClCa_ current in these cells (Figure 1c, Appendix A and Appendix A). Next, we addressed the role of endothelial TMEM16A in the tone of pulmonary arteries ex vivo. Isometric tension measurements of mouse pulmonary arteries, performed using wire myography, showed that Bbr induces a strong dose-dependent relaxation of mouse intrapulmonary arteries pre-constricted with 300 nM U46619 (Figure 1d). Vessels lacking endothelium (no EC) or incubated with 300 µm N_ω_-Nitro-L-arginine methyl ester hydrochloride (L-NAME) to inhibit nitric oxide production showed an attenuated vasorelaxation to Bbr (Figure 1d,e and Appendix A). These results suggest that endothelial TMEM16A contributes to the pulmonary arterial tone.

### 3.2. TMEM16A Accounts for Increased Ca^2+^-Activated Cl^−^ Current in IPAH PAECs

Next, we investigated TMEM16A in IPAH, a disease with an established endothelial dysfunction. First, we verified the presence of TMEM16A in von Willebrand Factor positive (vWF^+^) cells from healthy donors and IPAH patients employing immunofluorescence staining in 3D precision-cut lung slices (PCLS), as well as lung sections and PAECs (Figure 2a–c, Appendix A and Appendix A). We next sought to verify Ca^2+^-activated Cl^−^ current in IPAH PAECs employing whole-cell patch-clamp recordings. Our results showed significantly increased Bbr-sensitive current in IPAH PAECs in comparison to donor PAECs (Figure 2d,e and Appendix A).

### 3.3. Increased TMEM16A Disrupts Cl^−^ and Ca^2+^ Homeostasis of Human PAECs

To further elucidate if TMEM16A activity is causative for the development of endothelial dysfunction, subsequent experiments were performed in healthy primary human PAECs transduced with adenovirus Ano1^Ad^ encoding TMEM16A tagged with mCherry. As revealed by immunofluorescence and immunoblotting the TMEM16A expression was profoundly higher in primary human PAECs transduced with Ano1^Ad^ compared to mCherry encoding control adenovirus (Ctrl^Ad^) (Figure 3a, Appendix A and Appendix A). Whole-cell voltage clamp measurements showed increased Bbr-sensitive I_ClCa_ upon infection with Ano1^Ad^ highlighting the functional implication of TMEM16A overexpression (Figure 3b–d and Appendix A). Next, we addressed downstream events probably directed by elevated TMEM16A activity. Since elevated Cl^−^ conductance caused by increased TMEM16A activity resulted in resting membrane depolarization (Figure 4a), we used Ca^2+^ imaging to address its possible consequences on Ca^2+^ homeostasis. TMEM16A overexpression caused increased intracellular Ca^2+^ concentration [Ca^2+^]_i_ and in addition hinted at elevated store depletion (Figure 4b–d and Appendix A) as well as reduced extracellular replenishment mechanisms of PAECs (Figure 4e). In contrast, Ca^2+^ replenishment mechanisms of PASMCs overexpressing TMEM16A were enhanced (Appendix A).

### 3.4. Enhanced TMEM16A Activity Promotes Endothelial Dysfunction

TMEM16A-overexpressing primary human PAECs favor oxidative metabolism over glycolytic as demonstrated by higher oxygen consumption rate (OCR) to extracellular acidification rate (ECAR) ratio (OCR/ECAR) compared to Ctrl^Ad^ transduced cells (Figure 5a,b). While TMEM16A overexpression lowered PAEC proliferation, the Cas3/Cas7 activity as well as the cleavage of microtubule-associated protein 1A/1B-light chain 3 (LC3), remained unchanged suggesting that decreased cell number is not a direct consequence of increased apoptosis or autophagy (Figure 5c,d and Appendix A). Additionally, TMEM16A overexpression did not significantly alter the cell cycle of these cells (Figure 5e). Protein levels of several markers of proliferation (PCNA), apoptosis (cleaved PARP/PARP ratio) and cell cycle (Cyclin D1) were analyzed (Figure 5f and Appendix A).

One of the fundamental capabilities of endothelial cells is formation into more organized structures. Since enhanced activity of TMEM16A reduced proliferation with cells favoring oxidative, instead of angiogenesis-supporting glycolytic metabolism, we looked further into their angiogenic potential. In matrigel, healthy primary human PAECs formed a network of tubular structures whereas TMEM16A-overexpressing PAECs showed reduced spreading potential and tube-formation capabilities, indicating that these cells have a deficiency in new vessel formation (Figure 6a–d). Therefore, we examined whether TMEM16A overexpression affects signaling pathways involved in endothelial tube formation. Our data show, that TMEM16A- overexpression in primary human PAECs attenuated ERK1/2 phosphorylation, without significantly affecting Akt, MAPK p38, or JNK pathways (Figure 7a,b).

### 3.5. Increased TMEM16A Disrupts Induced Activation of eNOS

A fundamental part of endothelial cell identity is its ability to produce nitric oxide (NO). Since TMEM16A-overexpressing PAECs resulted in dysfunctional vessel formation, we looked further into alterations of the NO pathway. We found that increased TMEM16A activity, unexpectedly, did not affect baseline NO level, but diminished acetylcholine (ACh)-induced NO production (Figure 8a and Appendix A). At a protein level, overexpression of TMEM16A- caused eNOS alterations portrayed as more active eNOS phosphorylation at Ser1177 and inhibitory phosphorylation at Thr495 following ACh stimulation (Figure 8b and Appendix A). At basal, noninduced level, TMEM16A-overexpression led to higher phosphorylation that persists after ACh stimulation and goes hand in hand with decreased Thr495 phosphorylation (Figure 8c). Thus, enhanced TMEM16A activity causes increased activation of eNOS, without the accompanying increase in NO production.

Since overexpression of TMEM16A seemed to weaken the effect of ACh on primary human PAECs, we tested if chronic decrease in intracellular chloride concentration ([Cl^−^]_i_), a possible consequence of an increased TMEM16A activity, causes perturbations in the eNOS activating pathway observed upon TMEM16A overexpression. Indeed, incubation of primary human PAECs with Cl^−^-reduced Ringer’s solution resulted in increased phosphorylation of activatory Ser1177 at baseline, as well as 15 min after ACh stimulation, a similar eNOS phosphorylation pattern as was seen in the case of TMEM16A overexpression (Appendix A and Appendix A).

### 3.6. TMEM16A Upregulation in Donor Pulmonary Arteries Reduces Acetylcholine-Induced Vasorelaxation

Since ACh-induced NO production was inherently disrupted in TMEM16A overexpressing PAECs, we assumed that there would be impairment of ACh-induced vasorelaxation in pulmonary arteries upon TMEM16A overexpression. Finally, we overexpressed TMEM16A using Ano1^Ad^ in human pulmonary arteries obtained from healthy donors in order to mimic IPAH (Figure 9a). Immunofluorescence staining of donor 3D, precision-cut lung slices (PCLS) attested to the overexpression in the vWF^+^ cells of the vessels (Figure 9b, Appendix A and Appendix A). Isometric tension measurements of IPAH-like donor pulmonary arteries demonstrated attenuated ACh-mediated vasorelaxation. (Figure 9c,d and Appendix A).

## 4. Discussion

Endothelial cells (ECs) express a bewildering diversity of different chloride channels responsible for a variety of cellular functions [24]. The Ca^2+^-activated Cl^−^ channel (CaCC) TMEM16A has been previously detected in the mitochondria in human pulmonary artery endothelial cells (PAECs) [19]. Our investigation is the first to report the presence of active TMEM16A in the plasma membrane of primary human PAECs. Previously, we have demonstrated the importance of TMEM16A in PASMCs and revealed the therapeutic potential of targeting TMEM16A to reverse pathophysiological signatures of PH in animal models [18]. With our current results we provide further insights into the role of TMEM16A for the physiologic function of the endothelium and the potential impact of elevated TMEM16A activity for pulmonary endothelial dysfunction, a hallmark of PH.

The Ca^2+^-activated Cl^−^ channel TMEM16A, encoded by the gene ANO1, was found in many excitable tissues such as smooth muscle, cardiac muscle and sensory neurons. The channel is activated by relatively low intracellular Ca^2+^ and, upon Ca^2+^ binding, an allosteric change induces an opening of the channel pore leading to conduction of an outwardly rectifying current. Although Ca^2+^-activated Cl^−^ current has been studied at the functional level for more than 30 years in different tissues, only a few studies have investigated its role in endothelium. Almost two decades ago, by employing 3D, precision-cut lung slices, we reported the postnatal development of the chloride current contribution to the membrane potential in rat PAECs in situ [25]. A recent report indicated a relationship between TMEM16A activity and oxidative stress in endothelial cells in the systemic circulation. Transgenic endothelium-specific overexpression of TMEM16A facilitated Nox2/p22phox expression and reactive oxygen species production leading to endothelial dysfunction in a mouse model [26]. Our study is however the first to show the functional presence of TMEM16A in the cell membrane of human PAECs using patch-clamp recordings. We demonstrate the relevance of this channel for the endothelial function ex vivo and establish TMEM16A as an important player in human pulmonary endothelium under physiological and pathophysiological conditions.

Under disease conditions, elevated TMEM16A in endothelial cells of the systemic circulation was linked to hypertension by showing that TMEM16A plays a specific and critical role in mediating the hemodynamic response to angiotensin II [26]. Our investigations provide evidence for the functional consequences of increased TMEM16A activity and demonstrate enhanced chloride current in the cell membrane of PAECs from IPAH patients. In order to examine the functional consequences of the elevated TMEM16A expression in endothelial cells, we successfully established a model system for subsequent investigations. Using an IPAH-mimicking enhancement of TMEM16A activity achieved by adenoviral overexpression, we detected depolarization of resting membrane potential and altered Ca^2+^ homeostasis in primary human PAECs. Depolarization was also observed in PASMCs, as outlined in the current and the previous study [18]. However, the depolarization in both of the cell-types have different physiological consequences. After the removal and reintroduction of extracellular Ca^2+^, the Ca^2+^ influx decreased in PAECs overexpressing TMEM16A and increased in PASMCs. The difference between the two cell types might be the result of a cell-type specific channelome. The absence of L-type Ca^2+^ channels in PAECs [27,28,29], which participate mainly in voltage-dependent Ca^2+^ influx in many other cell types including PASMCs, indicates an alternative mechanism in PAECs. Although we did not perform additional experiments, according to previous studies it is possible that store-operated Ca^2+^ entry (SOCE) plays a key role here [30,31]. The reduced Ca^2+^ influx in TMEM16A-overexpressing PAECs can be a hint of mitochondrial dysfunction in these cells [31]. By emptying the endoplasmic reticulum (ER) with BHQ, we detected that stores hold higher amounts of Ca^2+^ in TMEM16A-overexpressing PAECs. Increased cytosolic Ca^2+^ concentration due to Ca^2+^ leak from the ER reticulum has been described in the coronary ECs of diabetic mice [32]. Thus, with regard to the elevated intracellular Ca^2+^ concentration we speculate that TMEM16A-overexpressing PAECs may develop a similar ER Ca^2+^ leak contributing to the observed increased Ca^2+^ level. Enhanced ER Ca^2+^ leak activates SOCE, a newly reported activator of calcium-dependent chloride channels [33,34]. Furthermore, a recent study has shown that TMEM16A is directly activated by Ca^2+^ release from the ER via the IP_3_ receptors [35]. Thus, the increased intracellular Ca^2+^ level together with depolarization might provide a re-entrant loop for the activation of the TMEM16A in IPAH PAECs.

Mitochondria are recognized at the crossroads of many cellular functions. Our investigations reinforced our hypothesis that mitochondrial function is altered in TMEM16A-overexpressing PAECs, highlighted by the predominance of oxidative over glycolytic metabolism. As previously reported, endothelial cells rely primarily on glycolysis with in vitro glycolysis rates comparable to or even higher than in cancer cells [36]. One of the advantages of aerobic glycolysis in ECs is that lower oxidative phosphorylation generates less reactive oxygen species (ROS) [37]. Pulmonary hypertension is characterized by perturbations in metabolic pathways affecting NO generation, smooth muscle proliferation, migration and oxidative stress [3,38]. Our data points to a TMEM16A-overexpression-driven shift toward oxidative phosphorylation in PAEC. However, further work is needed to determine whether this metabolic alteration, intimately related to mitochondrial function, represents a functional feature of a PAEC subtype or an early step in the disease development.

Chloride movement across the cell plasma membrane has been suggested to play an important role in regulating a variety of physiological processes, including cell proliferation, apoptosis, migration or mucus secretion. Elevated TMEM16A activity results in increased chloride efflux in endothelial cells and thus causes the decrease of the intracellular chloride concentration [39]. Our data show that TMEM16A overexpression in primary human PAECs led to decreased proliferation, wound-healing and tubulogenesis, while apoptosis remained unaffected. The TMEM16A-induced endothelial dysfunction, shown as aberrant proliferation and tube formation, was ERK1/2-dependent as TMEM16A overexpression led to decreased ERK1/2 but not JNK, p38 or Akt phosphorylation. These findings are in line with recent studies indicating that the cell-specific role of elevated TMEM16A for migration, proliferation or angiogenesis depends on the cellular environment and it is specific for a particular cell type [34]. Accordingly, our results very likely reflect that the ERK1/2 pathway in human PAECs was specifically affected by TMEM16A overexpression, leading to reduced wound-healing and vessel formation.

We show that IPAH-mimicking TMEM16A overexpression alters eNOS function in human PAECs. eNOS acts as a master regulator of vascular tone through the generation of NO and dysregulation of eNOS is a hallmark of many cardiovascular diseases. We noticed that baseline as well as ACh-induced activatory eNOS phosphorylation were increased, however, NO bioavailability was decreased under stimulated conditions in TMEM16A overexpressing cells. Increase in the intracellular concentration of free calcium has been proposed to regulate eNOS phosphorylation and activity [40]. Consistent with this, the enhanced TMEM16A activity-mediated increase in cytosolic Ca^2+^ levels in our study could lead to increased phosphorylation at eNOS activation sites, including Ser1177 [41,42]. The discordance between eNOS activatory phosphorylation and NO availability as observed in the present study was also found in endothelial cells incubated with insulin or VEGF [41]. Several potential mechanisms such as eNOS uncoupling, or dimer disruption as well as NO scavenging by ROS, (possibly raised by augmented oxidative metabolism), may contribute to the decreased NO availability in TMEM16A-overexpressing PAECs [41,42,43]. Our findings imply that increased TMEM16A activity in PAECs interferes with background eNOS activity and the stable production of nitric oxide, essential components for maintaining low vascular tone in normoxic conditions [44].

There are some discrepancies between our findings and the previously published reports on TMEM16A in IPAH PAECs, as we did not detect an increased apoptosis or a channel activator-mediated attenuated p38 phosphorylation. However, most of the results in the previous study have been generated on rat lung microvascular ECs based on activated mitochondrial TMEM16A. Thus, the regional and species-specific heterogeneity might explain the differences. Furthermore, previous studies postulated several stages in PAEC transformation in PAH [45,46,47]. Accordingly, our data suggest that increased TMEM16A activity in the cell membrane represents one stage of the IPAH PAEC phenotype [48]. Moreover, similar to our findings, IPAH patients may have disrupted eNOS pathways [49,50]. Although eNOS is considered a beneficial factor in the pulmonary arteries, persistent eNOS activation may also induce PH in mice and humans through PKG nitration causing disturbed vasodilation [51]. Chronic hypoxic and monocrotaline rat models of PH are characterized by impaired ACh-mediated vasorelaxation, further supporting our hypothesis [52].

Our study has some limitations: applying a broad range of functional read-outs made it necessary to utilize freshly isolated pulmonary arteries, fresh precision-cut lung slices as well as primary human endothelial cells in cell culture in one study. It is possible that culturing causes phenotypical changes of primary cells and this may impair the conclusions drawn from experiments in intact vessels. However, this type of combination of different methods still provides the backbone for the majority of preclinical studies in the vascular system. The other drawback is, that we can’t exclude changes in the biophysical properties of the TMEM16A channel in IPAH PAECs. Further investigations on activation and on the Ca^2+^-binding of the channel are needed [53]. Finally, we found a marked overexpression of TMEM16A in IPAH PAEC and the respective whole-cell Ca^2+^-activated Cl^−^ current as well as the Bbr-responsive current but we did not investigate other readouts in the IPAH PAECs because the number of available cells was very much limited for high throughput read-outs. Therefore, we cannot provide direct evidence that silencing of TMEM16A would normalize the IPAH phenotype.

Within the scope of this work, we were able to extend the pathological footprint of TMEM16A overexpression beyond its effect on pulmonary arterial smooth muscle cell and show its role in endothelial dysfunction. We traced the vasorelaxation and angiogenic defects of TMEM16A overexpression and the decrease in NO production in PAECs and established the disease-associated increased TMEM16A activity in PAH.

## Figures and Tables

**Figure 1 cells-09-01984-f001:**
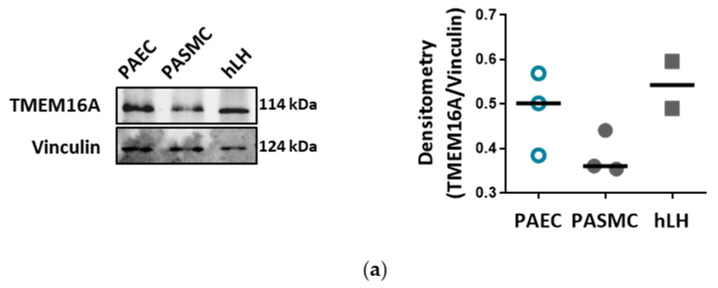
TMEM16A defines calcium-activated chloride current in pulmonary arterial endothelial cells (PAECs). (**a**) Western blot showing TMEM16A expression in donor PAECs, pulmonary arterial smooth muscle cells (PASMCs) and human lung homogenate (hLH) (N = 2–3 patients). Vinculin served as loading control. (**b**) Representative whole-cell, Ca^2+^-activated Cl^−^ current (I_ClCa_) traces (left) and normalized current-voltage (I-V) relationships measured with voltage clamp in PAECs showing the effect of benzbromarone (Bbr) (right). (**c**) Calculated Bbr-sensitive current in donor PAECs and PASMCs. Figures were generated with *n* = 5–13 cells from healthy donors. (**d**,**e**) Representative isometric tension measurements and quantification showing endothelial contribution of Bbr effectiveness on U46619 pre-constricted mouse pulmonary arteries with either endothelium removed (no EC; d) or incubation with 300 µm L-NAME (**e**) (*n* = 4–7). ** *p* < 0.01, *** *p* < 0.001, ANOVA with Bonferroni post-hoc test, data are presented as mean ± s.e.m.

**Figure 2 cells-09-01984-f002:**
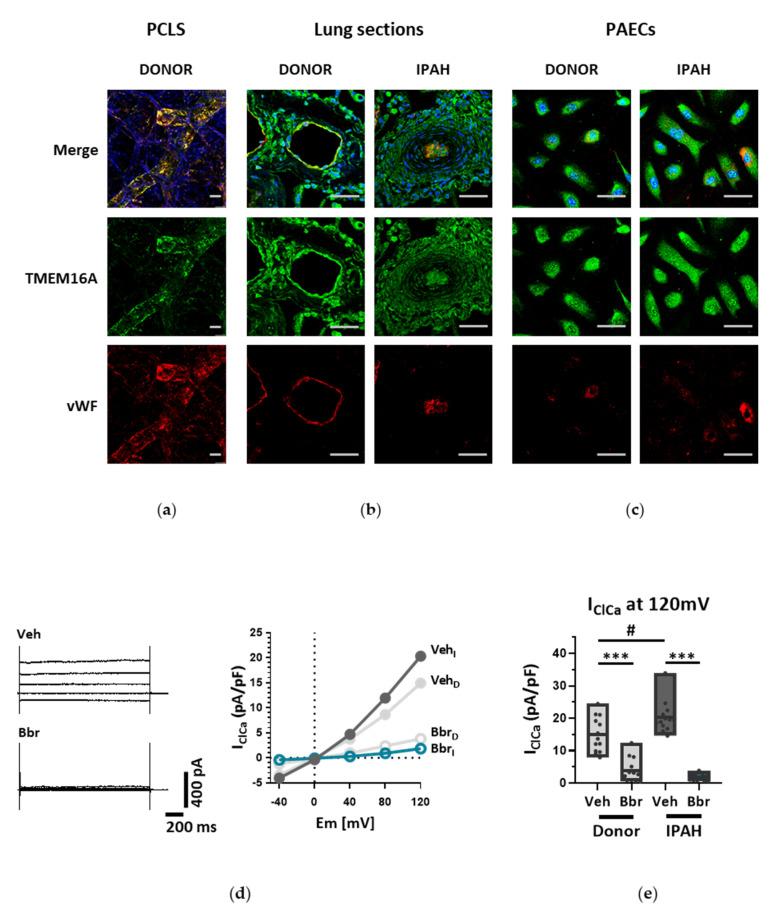
TMEM16A accounts for increased Ca^2+^-activated Cl^−^ current in IPAH PAECs. Immunofluorescence staining of (**a**) 3D precision cut lung slices (PCLS), (**b**) lung sections and (**c**) PAECs obtained from healthy donor lungs and patients suffering from IPAH (BP = antibody blocking peptide, scale bar = 50 µm for PCLS, 50 µm for PAECs and 50 µm for lung sections). (**d**) The effect of Bbr on representative whole-cell I_ClCa_ traces (left) and normalized current-voltage relationships (right) measured with voltage clamp in donor and IPAH PAECs (Veh_I_/Bbr_I_ = IPAH PAECs perfused with vehicle or Bbr; Veh_D_/Bbr_D_ = donor PAECs perfused with vehicle or Bbr). (**e**) Comparison of the Ca^2+^-activated Cl^−^ current density of donor and IPAH PAECs at 120 mV. Figures were generated with *n* = 12–13 cells from at least N = 4 patient samples. # *p* < 0.01, *** *p* < 0.0001, ANOVA with Bonferroni post-hoc test, data are presented as mean ± s.e.m.

**Figure 3 cells-09-01984-f003:**
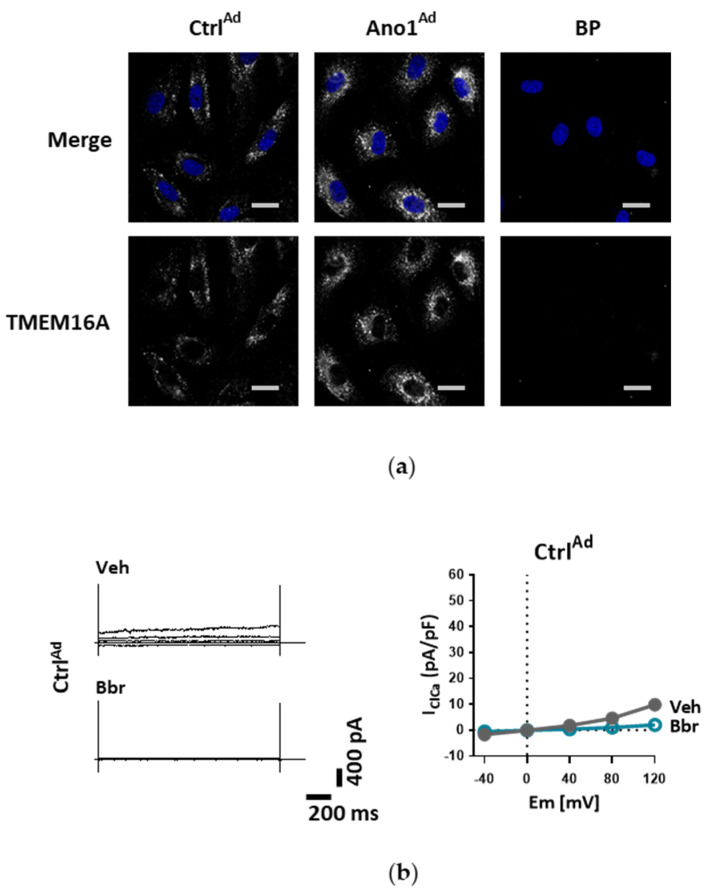
Upregulation of TMEM16A in human PAECs. (**a**) Immunofluorescence staining of TMEM16A in PAECs infected with Ctrl^Ad^ or Ano1^Ad^ (BP = antibody blocking peptide; scale bar = 20 µm). (**b**) Representative whole-cell I_ClCa_ traces (left) and normalized current-voltage relationships (right) measured with voltage-clamp showing the effect of Bbr in donor PAECs transfected with Ctrl^Ad^. (**c**) Representative whole-cell I_ClCa_ traces (left) and normalized current-voltage relationships (right) measured with voltage-clamp showing the effect of Bbr in donor PAECs transfected with Ano1^Ad^ and overexpressing TMEM16A. (**d**) Consecutive, calculated Bbr-sensitive current comparing primary PAECs infected with Ctrl^Ad^ or Ano1^Ad^. Figures were generated with 8–13 cells from N = 2 healthy donors, data are presented as mean ± s.e.m.

**Figure 4 cells-09-01984-f004:**
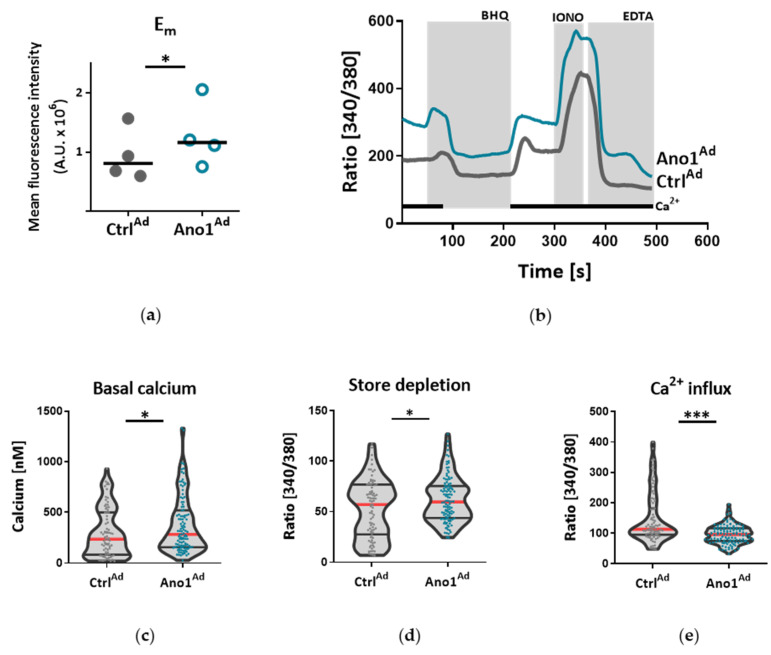
TMEM16A-mediated membrane depolarization disrupts Ca^2+^ dynamics of human PAECs. (**a**) Fluorometric measurements indicating shift in relative resting membrane potential (E_m_) of donor PAECs infected with Ctrl^Ad^ or Ano1^Ad^ using DiBAC_4_(3) dye. (**b**) Representative traces depict changes in intracellular Ca^2+^ detected in PAECs transfected with Ctrl^Ad^ or Ano1^Ad^. (**c**–**e**) The effect of TMEM16A overexpression on cytosolic baseline Ca^2+^ concentration ([Ca^2+^]_i_), store depletion and Ca^2+^ influx using Fura-2 in donor PAECs infected with Ctrl^Ad^ or Ano1^Ad^ (BHQ = butylhydroquinone). Figures were generated with 80-116 cells from N = 3 healthy donors. * *p* < 0.05, *** *p* < 0.001, paired (**a**) and unpaired t-tests (**c**–**e**).

**Figure 5 cells-09-01984-f005:**
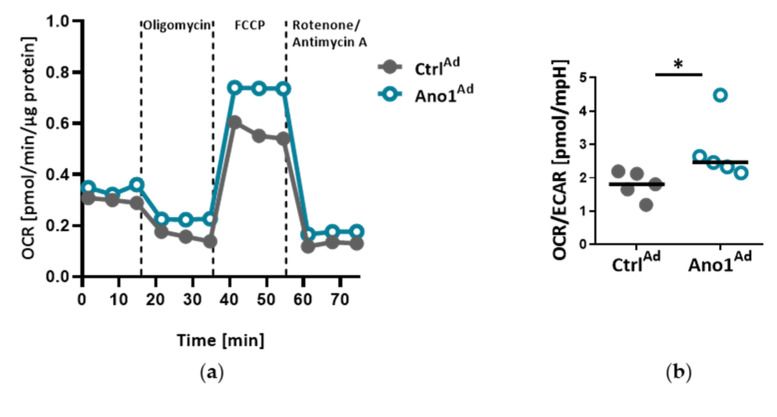
Enhanced TMEM16A mediates metabolic changes of PAECs. (**a**–**b**) Seahorse mitochondrial stress test profiles of TMEM16A-overexpressing primary PAECs showing the ratio of oxygen consumption rate (OCR) to extracellular acidification rate (ECAR) OCR/ECAR. (**c**) Proliferation of human PAECs overexpressing TMEM16A measured with thymidine incorporation (*n* = 5). (**d**,**e**) Cas3/Cas7 apoptosis assay and cell-cycle analysis of human PAECs overexpressing TMEM16A. (STS = staurosporin). (**f**) Western blots of PAECs infected with TMEM16A-overexpressing Ano1^Ad^ or control Ctrl^Ad^ with quantifications of PCNA, cleaved PARP/PARP and Cyclin D1. Figures were generated with 13 separate sets of experiments. * *p* < 0.05, ratio-paired (**a**) or paired t-test.

**Figure 6 cells-09-01984-f006:**
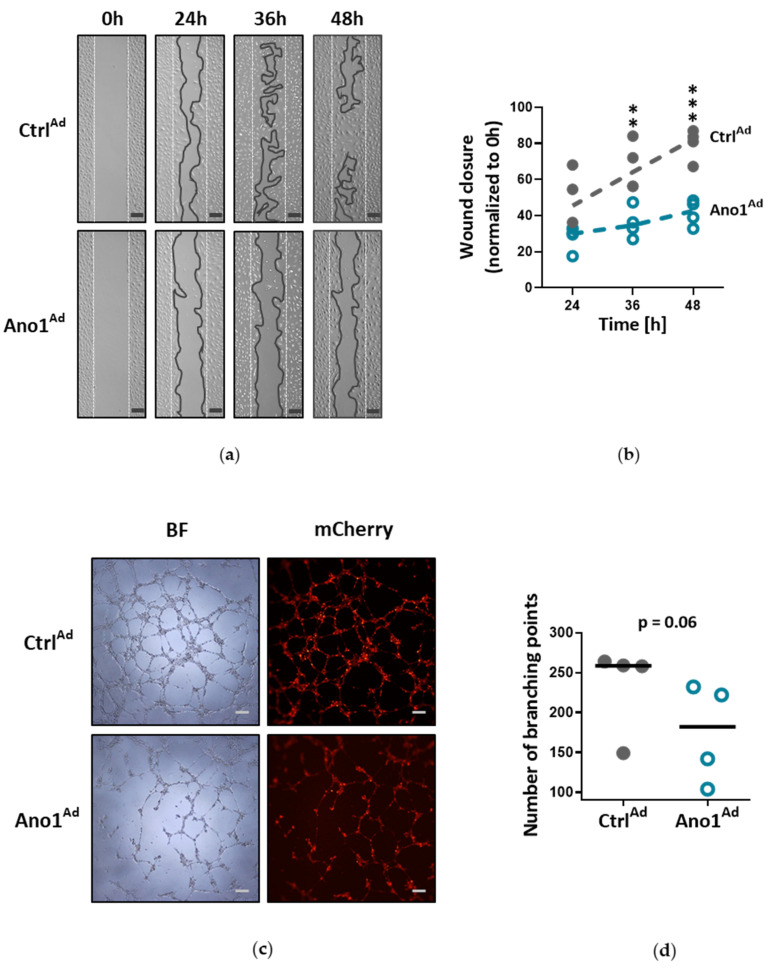
Increased TMEM16A activity causes angiogenic dysfunction. (**a**) Representative pictures of PAECs wound healing assay, expressing either Ano1^Ad^ or control Ctrl^Ad^, taken after 24, 36 and 48 h (scale bar = 200 µm). (**b**) Quantification of the wound healing assay. (**c**) Matrigel tube-formation assay with representative bright-field (BF) and mCherry fluorescence pictures of Ano1^Ad^ or Ctrl^Ad^-infected PAECs (scale bar = 200 µm). (**d**) Quantification showing the number of branching points in comparison to Ctrl^Ad^. Figures were generated with 4 separate sets of experiments with triplicate in each group. ** *p* < 0.01, *** *p* < 0.001, (**b**) ANOVA with Bonferroni post-hoc test, (**d**) paired t-test.

**Figure 7 cells-09-01984-f007:**
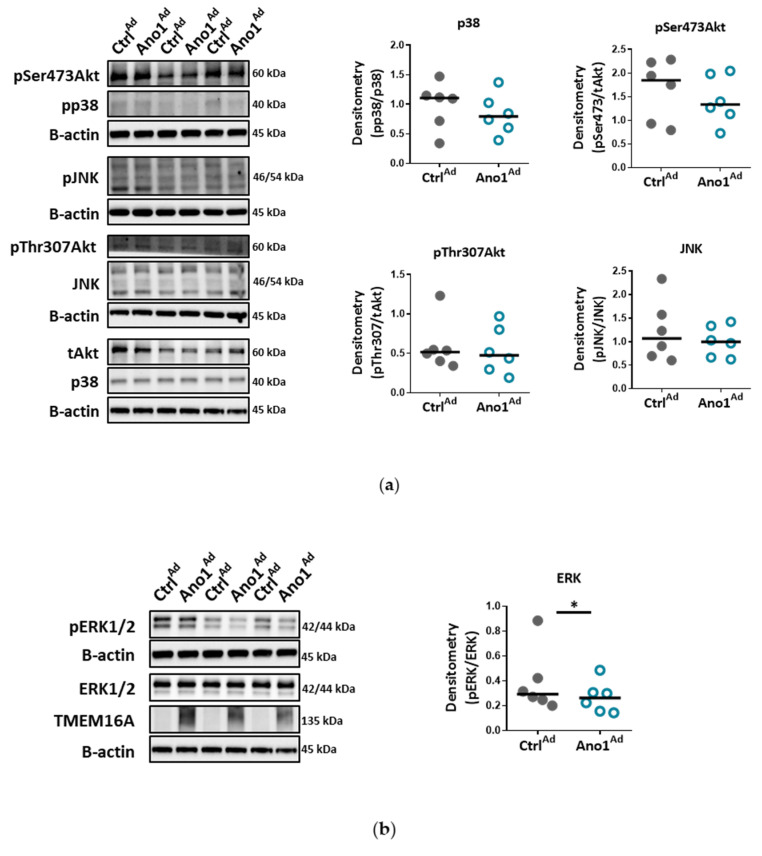
Elevated TMEM16A activity alters ERK1/2 signaling. (**a**) Western blots showing protein levels of p38, Akt and SAPK/JNK obtained from of PAECs infected with either Ano1^Ad^ or control Ctrl^Ad^. Figures were generated with 6 samples. (**b**) Western blots showing ERK1/2 pathway activation in PAECs infected with either Ano1^Ad^ or Ctrl^Ad^ with quantification. Figures were generated with 6 samples. * *p* < 0.05, ratio-paired t-test.

**Figure 8 cells-09-01984-f008:**
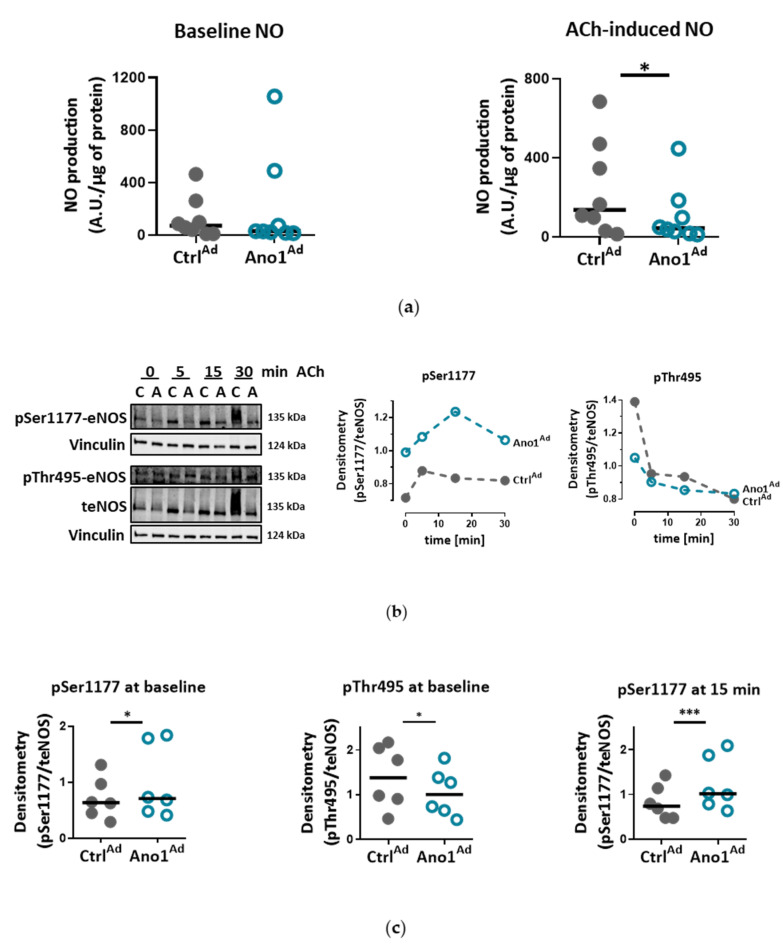
Elevated TMEM16A activity disturbs eNOS activation: (**a**) Noninduced nitric oxide levels and ACh-induced nitric oxide production of control (Ctrl^Ad^) and TMEM16A-overexpressing (Ano1^Ad^) human PAECs. Figures were generated with 8 sets of experiments with quadruplicate in each group and normalized to protein content. (**b**) Western blots showing ACh-induced changes in eNOS phosphorylation of Ctrl^Ad^ and Ano1^Ad^-infected donor PAECs with quantification following the eNOS phosphorylation pattern at activatory Ser1177 and inhibitory Thr495 sites after 5, 15 and 30 min of ACh stimulation. (**c**) Quantification of basal, noninduced level of eNOS phosphorylation at Ser1177 and Thr495 as well as phosphorylation of Ser1177 15 min after ACh stimulation. Figures were generated with 6 samples. * *p* < 0.05, *** *p* < 0.001, ratio-paired t-test.

**Figure 9 cells-09-01984-f009:**
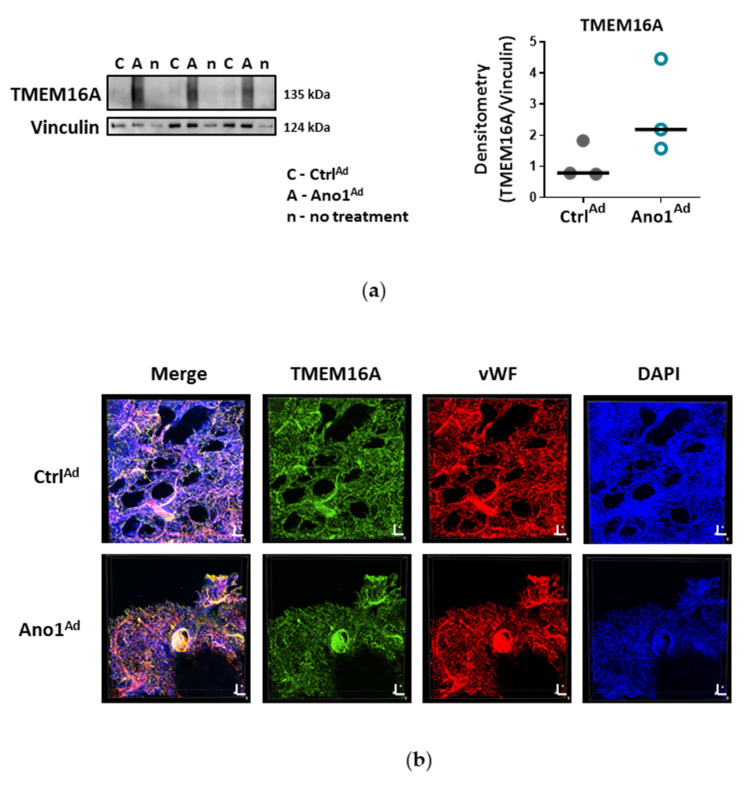
Enhanced TMEM16A results in reduced ACh-induced vasorelaxation of human donor pulmonary arteries. (**a**) Western blot of healthy donor pulmonary artery infected with either Ctrl^Ad^ or Ano1^Ad^ with quantification of TMEM16A overexpression (right). (**b**) Immunofluorescence staining of donor 3D, precision-cut lung slices (PCLSs) infected either with Ctrl^Ad^ or Ano1^Ad^ (scale bar = Width = 1257.93 µm, Height = 1257.93 µm, Depth = 82.00 µm for Ctrl^Ad^ and 247.00 µm for Ano1^Ad^). (**c**) Representative isometric tension measurements showing ACh-mediated vasorelaxation of U46619 pre-constricted donor pulmonary arteries (PAs) infected with either Ctrl^Ad^ or Ano1^Ad^ (left) and quantification of the results (right) showing 2–3 PAs from the same donor in both groups. (**d**) The effect of TMEM16A overexpression on 30 µM ACh-induced vasorelaxation of 300 nM U46619 preconstricted healthy donor pulmonary arteries. Figure was generated with 7 pairs of vessels taken from N = 3 donors. Unpaired t-test.

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
