# Peer review of "Endothelial Dysfunction Following Enhanced TMEM16A Activity in Human Pulmonary Arteries"

_cells, 2020, doi:10.3390/cells9091984_

Round 1

Reviewer 1 Report

This manuscript by Mauer et al. examined the potential pathological link between CaCC/TMEM16A activity and endothelial dysfunction in human pulmonary artery endothelial cells (PAEC). The authors verified the expression of TMEM16A in human PAECs. Using the patch-clamp technique in the whole-cell voltage-clamp configuration, the authors detected Bbr-sensitive currents in human PAECs. Furthermore, the authors observed a significant increase in the current density of these Bbr-sensitive currents in PAECs harvested from patients with IPAH compared to donor controls. Overexpression of TMEM16A using an adenoviral expression system in donor PAEC allowed the authors to examine how TMEM16A activity contributes to endothelial cell dysfunction. The authors found that overexpression of TMEM16A altered Ca2+ homeostasis, mitochondrial function, ERK1/2 signaling, and eNOS function. Additionally, the authors reported that overexpression of TMEM16A in donor PAEC led to decreased cell proliferation, migration, and angiogenesis.

This is a well-written manuscript that provides insight into the role of CaCC/TMEM16A in endothelial dysfunction, an important hallmark in the pathogenesis of pulmonary hypertension.  These findings are novel as the authors are the first to show that overexpression of TMEM16A in PAEC alters Ca2+ handling, angiogenesis, ERK1/2 signaling, and eNOS activity.  This study presents some intriguing observations, however, there are several concerns that the authors should consider.

Specific comments:

  • This is a nice study providing compelling evidence that TMEM16A is sufficient for altering several physiological processes and key signaling pathways involved in the development of endothelial dysfunction. However, the authors should consider including a series of experiments to demonstrate the necessity of TMEM16A activity in endothelial dysfunction. It would be interesting to see if silencing TMEM16A expression in IPAH PAEC could rescue normal endothelial function.  

Reviewer 2 Report

This is a generally well-designed study assessing the role of TMEM16A in a broad range of responses in cells from blood vessels of the lung. The assessment of proliferative, cell signaling, and contractile pathways benefits from the use of human cells enhancing the translational potential of this work. However, further clarification of key details is necessary to maximize the potential of this study.

Broad Comments:

  • While the assessment of wide array of effects of TMEM16A in pulmonary arteries is an underreported area of research and represents an extensive and valuable effort by the authors, the breadth of responses assessed somewhat limits the mechanistic insight provided.
  • The use of human cells and arteries enhances the translational impact of the work.
  • Many values are expressed as % of control. Not reporting raw numerical values limits critical assessment of data collection (are values in line with those previously reported) and can artificially enhance detection of statistically significant differences.
  • Writing can be improved to provide greater clarity and consistency throughout the manuscript (see specifics in minor comments below).
  • While cell culture is a valuable tool for addressing responses, phenotype can also change in cultured systems. While vasorelaxation responses provide complimentary approach for effects on vascular tone, studies examining cell proliferation and signaling are not paired to data in intact vessels. Some mention of this limitation should be included.

Specific Comments:

Major concerns:

  • Mention that purchased PAECs are from humans and that patient characteristics are included in supplement should be included in methods (page 3, line 96). Same applies to PASMCs (line 102). Clear indication of supplementary materials needs to be addressed within the main text. This also applies to the antibody table; should be mentioned at the location of the first antibody used.
  • The concentration of U46619 used to contract vessels for isometric tension measurements is not mentioned in the methods or results section. Furthermore, the amount of contraction elicited by U46619 needs to be included for reference. Additionally, the term “vasodilation” is not appropriate for tension measurements and the term “vasorelaxation” should be used instead.
  • The authors associate increased TMEM16A activity with expression. Does PAH also change channels properties such as Ca2+/calmodulin sensitivity? This may be important as most experiments in the study only recapitulate enhanced expression as a model for study.
  • The effects of EC removal or L-NAME inhibition on Brb vasorelaxation (figure 1), although statistically significant, are relatively minor compared to the total vasodilatory response. The majority appears to be mediated by SMC TMEM16A. Also the experiment assesses the contribution of endothelial TMEM16A to total relaxation by TMEM16A in preconstricted vessels. Therefore, the statement (page 8, lines 360-361) indicating that EC TMEM16A contributes significantly to the control of PA tone is an overinterpretation of data. Also applies to statement on line 377. Assessment of endothelium-dependent dilation in EC specific TMEM16A knockout mouse could be utilized to make this statement.
  • Supplemental Figure 3c: Why is there no TMEM16A expression in the control adenovirus PAECs and SMCs? It was detectable in Figure 1a. While the ano1Ad should increase the expression, there should be some expression. Same applies to Figure 5.
  • While the use of mouse vessels is described in the methods, there use in experiments is unclear. The whole artery experiments which use mice need to be clarified in the results.
  • Scale bars are necessary for figures currently lacking them.

Minor concerns:

  • English could be improved/clarified in several cases. For instance, line 56, the word choice of “underlining” seems improper. Furthermore, there are a number of editorial issues including subscripting CO2 in cull culture methods and adding a space between # and hour (h) for incubations. Also spacing of CHAPS [2] buffer is inconsistent.
  • Line 112: the term steaks usually only applies to muscularized tissue. Perhaps an alternative word would be more accurate.
  • The abbreviation precision-cut lung slices (PCLS) is not clearly identified in the text (page 3, lines 111 and 126).
  • Inclusion of specific [Cl-] in numerical form (page 5, lines 187-190) would be beneficial to the reader.
  • Figure 1, d and e. Axis label should read % relaxation and numbers should be positive (+). Negative relaxation would correspond to contraction.
  • Figure 1 legend: the corresponding letters for L-NAME and endothelium removal (d and e) are backwards in the legend.
  • Supplemental Figure 2. NC needs to be defined.
  • Line 441: “ration” should be “ratio”.
  • Figure 5 legend: Define STS here as well as supplemental legend.
  • Ach (for acetylcholine) should be ACh throughout. Should also be defined at first use (line 186) and be used consistently throughout, not intermittently switching between full version and abbreviation.
  • Figure 8 legend. b does not show changes in ERK1/2 yet is stated in the legend.
  • Figure 9. It appears as if adenovirus figures are repeated in miniature.

Round 2

Reviewer 1 Report

The authors have adequately and appropriately addressed my concern.

Reviewer 2 Report

The present study highlights the role of TMEM16A in endothelial dysfunction in human pulmonary hypertension. This revised manuscript has improved the clarity and presentation of key methodological descriptions and data presentation and interpretation to highlight the role of TMEM16A in a broad range of responses in pulmonary artery smooth muscle and endothelial cells.